# Obesity Surgery Improves Hypogonadism and Sexual Function in Men without Effects in Sperm Quality

**DOI:** 10.3390/jcm11175126

**Published:** 2022-08-31

**Authors:** Inka Miñambres, Helena Sardà, Eulalia Urgell, Idoia Genua, Analía Ramos, Sonia Fernández-Ananin, Carmen Balagué, Jose Luis Sánchez-Quesada, Lluís Bassas, Antonio Pérez

**Affiliations:** 1Endocrinology and Nutrition Department, Hospital de la Santa Creu i Sant Pau, 08041 Barcelona, Spain; 2Medicine Department, Universitat Autònoma de Barcelona (UAB), 08193 Barcelona, Spain; 3CIBER of Diabetes and Metabolic Diseases (CIBERDEM), 08041 Barcelona, Spain; 4Biochemistry Department, Hospital de la Santa Creu i Sant Pau, 08041 Barcelona, Spain; 5Endocrinology and Nutrition Department, Hospital Germans Trias i Pujol, 08916 Badalona, Spain; 6General Surgery Department, Hospital de la Santa Creu i Sant Pau, 08041 Barcelona, Spain; 7Cardiovascular Biochemistry Group, Research Institute of the Hospital de la Santa Creu i Sant Pau (IIB Sant Pau), 08041 Barcelona, Spain; 8Molecular Biology and Biochemistry Department, Faculty of Medicine, Universitat Autònoma de Barcelona (UAB), 08193 Barcelona, Spain; 9Andrology Department, Fundació Puigvert, 08025 Barcelona, Spain

**Keywords:** hypogonadism, obesity, obesity surgery, sperm, fertility

## Abstract

(1) Background: Obesity is associated with hypogonadism, sexual dysfunction, and impaired fertility in men. However, its effects on semen parameters or sexual function remain debatable. (2) Methods: This paper involves a longitudinal study in men submitted for obesity surgery at a university tertiary hospital. Patients were studied at baseline and at 6, 12, and 18 months after obesity surgery. At each visit, anthropometry measures were collected and hormonal and semen parameters were studied. Sexual function was evaluated with the International Index of Erectile Function (IIEF). (3) Results: A total of 12 patients were included. The average body mass index of patients decreased from 42.37 ± 4.44 to 29.6 ± 3.77 kg/m^2^ at 18 months after surgery (*p* < 0.05). Hormonal parameters improved after obesity surgery. The proportion of sperm cells with normal morphology tended to decrease from baseline and became most significant at 18 months (5.83 ± 4.50 vs. 2.82 ± 2.08). No significant changes were found in the remaining semen parameters. Erectile function improved significantly at six months after surgery. (4) Conclusions: The authors believe that, in general, the effects of obesity surgery on fertility may be limited or even deleterious (at least in the short and midterm follow-up).

## 1. Introduction

Obesity is an increasingly prevalent worldwide disease which is reaching epidemic proportions [1] and confers an increased mortality due to its related complications, such as multiple metabolic and cardiovascular disorders [2]. It is also linked to other health concerns including reproductive health [3].

Hypogonadism is more prevalent in men with obesity and metabolic syndrome, since testosterone levels appear to be inversely related to body mass index (BMI) and with the presence of obesity-associated comorbid conditions such as type 2 diabetes, obstructive sleep apnea, and metabolic syndrome [4,5,6,7]. Data from the European male ageing study (EMAS) have shown that increased BMI was associated with decreased total and free testosterone concentrations and with normal or decreased concentrations of luteinizing hormone (LH) [8]. This has suggested the presence of a partial functional hypogonadotropic hypogonadism, the so-called ‘male obesity-related secondary hypogonadism’ (MOSH) [8,9]. The altered hormonal regulation seen in obesity seems to present along with a greater degree of sexual disfunction when compared with normal weight controls [10]. Additionally, obesity has been linked with a decrease of sperm quality and, therefore, male fertility [11,12]. Although the proposed causal pathophysiological mechanisms involved in this are not well understood, underlying oxidative stress and inflammation may be important implicated mechanisms [13].

Lifestyle modifications that lead to weight loss have been determined to improve most obesity-related metabolic comorbidities and to increase total and free testosterone concentrations in men [14,15,16,17]. Bariatric surgery is currently the most effective treatment for morbid obesity, resulting in a significant and permanent weight loss. Concerning gonadal function, surgery has been described to achieve an 87% resolution of the male obesity-related secondary hypogonadism [3]. However, the effects of obesity surgery on semen parameters or sexual function remain debatable.

The aim of the present study was to analyze the effects of bariatric surgery on hormonal and semen parameters and sexual function in men with severe obesity.

## 2. Materials and Methods

This was a longitudinal study performed at a university tertiary hospital in Spain. The study included twelve men who met the National Institute of Health criteria for obesity surgery and were already scheduled for surgical procedures. All subjects gave written informed consent to participate in the present study. The protocol was approved by the clinical research ethics committee of the Hospital de la Santa Creu i Sant Pau. All procedures performed involving human participants were in accordance with the 1964 Helsinki declaration and its later amendments or comparable ethical standards.

We excluded patients above 55 years old and patients with kidney, liver, gonadal, or pituitary disease or concomitant treatments that could disturb/interfere with sexual hormone concentrations or semen parameters. All patients included were studied before (baseline) and at 6, 12, and 18 months after bariatric surgery.

Anthropometry parameters (weight, height, and BMI) and the presence of comorbidities associated with obesity were recorded before surgery and during follow-up. Dyslipidemia was defined as the presence of any of the following: triglyceride concentrations ≥ 1.7 mmol/L, HDLc concentrations < 1 mmol/L, LDLc concentrations > 4.14 mmol/L, or hypolipidemic treatment. The presence of metabolic associated fatty liver disease (MAFLD) was assessed by means of a baseline abdominal echography and by the NAFLD fibrosis score at baseline and follow-up [18]. The percentage of total weight loss (%TWL) was defined as the preoperative weight minus the follow-up weight, divided by the preoperative weight and multiplied by 100 [19].

Hormonal parameters measured included total testosterone, sex hormone binding globulin (SHBG), estradiol, prolactin, luteinizing hormone (LH), and follicle-stimulating hormone (FSH). Total testosterone and prolactin were determined using the electrochemiluminescence immunoassay (Cobas E601 analyzer; Roche Diagnostics GmbH, Mannheim, Germany). Assessment of FSH, LH, and estradiol was performed with the chemiluminescent microparticle immunoassay (CMIA) (Architect; Abbott Laboratories, Chicago, IL, USA), and SHBG was measured using the chemiluminescent enzyme immunoassay (Immulite 1000; Siemens Healthcare Diagnostics, Llanberis, UK). We calculated free testosterone by Vermeulen’s equation [20]. We defined hypogonadism as a total testosterone value < 9.2 nmol/L [21].

Semen samples were collected at each study visit on the premises of the hospital by masturbation in a clean, non-toxic container. These samples were immediately delivered to the laboratory. A sexual abstinence of three to five days was required. After liquefaction at 37 °C (98.6 °F) under gentle mixing, semen analysis was performed according to methods and reference limits of the WHO [22,23]. Semen volume was measured by recording the net weight of the container, assuming a density of 1 g/mL. pH was assessed with colorimetric strips; sperm concentration was performed on diluted, immobilized samples using hemocytometer chambers; motility was analyzed at room temperature; and sperm morphology of fixed smears stained with Diff-Quik was carried out following WHO criteria, including calculation of teratozoospermia index (TZI) [22]. Sperm DNA fragmentation was analyzed with the sperm chromatin dispersion test (SCD) [24]. Anti-sperm antibodies were identified with the mixed antiglobulin reaction (MAR) test (SpermMar IgG, Fertipro, Beermen, Belgium). The World Health Organization’s reference values for human semen characteristics were used when needed [23].

Sexual function was evaluated with the International Index of Erectile Function (IIEF). The questionnaire includes five main domains of male sexual function: erectile function, orgasmic function, sexual desire, intercourse satisfaction, and overall satisfaction. A score of 0/1–5 is awarded to each of the 15 questions of the test. Higher scores indicate more conserved sexual function. The total possible score for the erectile function domain ranges from 1 to 30, and erectile dysfunction is classified into four categories based on the scores: severe (6–10), moderate (11–16), mild (17–25), and no erectile dysfunction (26–30) [25].

A minimum sample size was calculated considering clinically significant changes in sperm and testosterone concentrations. In both cases, an alfa error = 0.05 and beta = 0.20 were considered. By assuming a clinically relevant mean difference in sperm concentrations of 15 million/mL (pre- and post-surgery) and a standard deviation of 15 million/mL, the resulting sample size was eight patients. When considering an expected change in total testosterone concentrations of 2.5 nmol/L with a 2 nmol/L standard deviation, the resulting sample size was six patients.

The statistical analyses were made with the SPSS 26.0 program. Variables were expressed as means ± standard deviation (SD) for continuous variables and absolute numbers with percentages for categorical variables. We used non-parametric tests (i.e., the Wilcoxon and McNemar tests) to analyze the continuous and categorical variables during follow-up, respectively. Correlations between changes observed were performed by means of Spearman correlation analysis.

## 3. Results

A total of 12 male patients completed the follow-up. Baseline and final clinical characteristics are shown in Table 1. Seven patients (58.3%) underwent gastric bypass and five (41.7%) underwent sleeve gastrectomy. The evolution of BMI and the percentage of total weight lost (%TWL) during the follow-up are shown in Figure 1.

The hormone profile evolution during follow-up is shown in Table 2. Hypogonadism was present in four patients (33.3%) at baseline and resolved after surgery in all cases. Total and free testosterone in patients with and without hypogonadism were 7.72 ± 1.53 vs. 13.89 ± 4.86 nmol/L (*p* = 0.036) for total testosterone and 0.22 ± 0.04 vs. 0.31 ± 0.05 nmol/L (*p* = 0.009) for free testosterone, respectively. The increase observed in these hormonal concentrations and in SHGB after bariatric surgery, according to the presence or absence of hypogonadism, was similar (data not shown). There was a significant positive correlation between the increase in total testosterone and the percentage of total weight lost at 18 months (rs 0.782; *p* = 0.004). Changes in SHBG at 18 months was also correlated with the percentage of total weight lost (rs 0.718; *p* = 0.013) and marginally correlated with changes in NAFLD fibrosis score (rs −0.6; *p* = 0.051).

Regarding semen parameters, there was a trend towards a decrease in the percentage of sperm cells with normal morphology after bariatric surgery (reaching statistical significance at 18 months) and an increase in the number of sperm with multiple defects (12 months). No significant changes were found in the rest of semen parameters after bariatric surgery (Table 2). The proportion of patients with one or more abnormal parameters was not reduced after the surgery. The increase in the percentage of sperm with multiple abnormalities seen at 12 months correlated with %TWL (rs 0.886; *p* = 0.019). None of the patients showed the presence of anti-sperm antibodies throughout the study.

Concerning sexual function, erectile function assessed by means of the IIEF test improved significantly at six months after surgery. All scores from the IIEF test are shown in Table 3.

## 4. Discussion

Our study confirms that weight loss attained after bariatric surgery increases total testosterone, free testosterone, and SHBG, resulting in the complete resolution of MOSH in severe obese men. The results demonstrate improved sexual function without an impact on sperm concentration and motility and an overall decline of morphology over time.

Our study shows that 33.3% of male patients with severe obesity submitted to bariatric surgery had presurgical testosterone concentrations in the range of hypogonadism (<9.2 nmol/L) [21]. After surgery, they all recovered from hypogonadism and the whole group experienced a significant increase in total and free testosterone and FSH concentrations, which indicates an improvement of the hormonal pattern characteristic of the obesity-related secondary hypogonadism (MOSH). Etiology of MOSH may be multifactorial and includes not only the diminished SHBG concentrations seen in obesity but also an increased aromatase activity, as well as the production of adipocytokines and gut derived endotoxins that impair kisspeptin signaling in the hypothalamus (and, consequently, GnRH secretion) [9,26]. In this respect, in our study we found increased SHBG concentrations as well as increased free testosterone concentrations after bariatric surgery, which highlights the fact that not all changes observed in testosterone concentrations are driven by the effects of obesity and weight loss on SHBG concentrations. Our findings are in agreement with the data of previous studies analyzing the effects of bariatric surgery on testosterone concentrations in men [3,27]. Previous studies investigating the effects of diet showed how an increase in total and free testosterone concentrations is correlated with the extent of weight loss. A metanalysis found that a diet-induced weight loss of 9.8% was associated with a significant increase in total testosterone of 2.8 nmol/L. Another metanalysis, which included a total of 567 patients, found that each 5 kg of weight reduction increased testosterone by 1 nmol/L [16,17]. In our study, although the small number of patients included may hinder the achievement of significant associations, we also found a correlation between weight loss and the improvement in testosterone concentrations. Therefore, weight loss can be described as the mainstay of MOSH treatment and bariatric surgery as the most effective way of achieving its resolution.

Sexual function was assessed in our study by means of the IIEF questionnaire, which estimates erectile function, orgasmic function, sexual desire, intercourse satisfaction, and overall satisfaction [25]. Our results show that bariatric surgery improved erectile function with no relevant effects on the other domains of sexual functionality. The effects of bariatric surgery on erectile function have been much less studied that the effects on hormonal concentrations, but our data are consistent with the few previous studies published that confirm a positive effect of bariatric surgery on IIEF-5 [28].

Despite the improvement in testosterone concentrations and sexual function, we did not observe a parallel improvement in sperm quality at any time point studied. In fact, the number of morphologically normal sperm was found to decrease after bariatric surgery, which corresponded with the increase found in the number of sperm with multiple abnormalities. This finding is intriguing, considering the significant improvement in the hormonal milieu of the obesity secondary hypogonadism. Previous studies analyzing the effects of massive weight loss with bariatric surgery are scarce. Initially, small case series pointed towards a deleterious effect of obesity surgery on fertility in men [29,30,31]. These anecdotic cases have been followed by a few contradictory prospective studies including small numbers of subjects. El Bardisi et al. [32] investigated the effects of sleeve gastrectomy on semen parameters at 12 months after surgery and found that men with previous oligo- and azoospermia increased their sperm concentrations. Samavat et al. [33] also found an improvement in seminal volume and viability at six months after surgery. However, these encouraging results have not been confirmed thereafter. Legro et al. [34] and Reis et al. [35] did not find any change in sperm parameters at 12 and 24 months after surgery, respectively, whereas three recent studies have reported a decrease in sperm concentration and total sperm count [36,37] and sperm volume [38]. Carette et al. [36] also found a decrease in the number of morphological normal sperm 12 months after obesity surgery. A recent meta-analysis summarizes the available literature concerning the effects of bariatric surgery in semen characteristics and concludes that this has no effect on sperm volume, concentration, total count, morphology, motility, or viability [39]. Conversely, information concerning the effect of weight loss through lifestyle modifications on sperm parameters is scarce but seems to show an improvement in some parameters of seminal quality such as sperm concentration, sperm count, or sperm morphology [40,41,42]. However, this information should be interpreted with caution due to the small number of studies available and the fact that there is a lack of head-to-head comparisons between lifestyle interventions and obesity surgery. In our study, we did not find any effect of bariatric surgery at any time studied on total sperm count or sperm concentrations. However, our findings which show a decrease in the percentage of normal sperm, together with an increase in the number of sperms with multiple abnormalities after obesity surgery, may indicate that obesity surgery can impair male fertility (at least during a period of weight loss). It is important to note, however, that the patients included in the present study (as well as in most previous studies) had normal seminal parameters on average, and that the small sample size of the present study as well as of previous studies limits drawing strong conclusions. This fact may have hindered the finding of potential changes in seminal quality and warrants future studies in males with obesity and seminal impairment at baseline. As a complement of basic sperm analysis, we analyzed DNA fragmentation in our study, finding an increase in DNA fragmentation during follow-up that did not reach statistical significance. Only three previous studies have analyzed DNA fragmentation, finding either no changes [33,37] or a decrease in this index [36].

While the factors that mediate the changes observed in gonadal hormones may involve the reversal of the previously commented mechanisms involved in the pathogenesis of MOSH, the factors leading the observed changes in semen parameters are less clear. Possible explanations for a worsening of semen characteristics after obesity surgery may be the decreased absorption of nutrients, the presence of nutritional deficiencies, and the possible release of toxins from adipose tissue [43,44]. It should be noted that patients submitted to obesity surgery at our institution are prescribed daily multivitamins and are examined for vitamin D, iron, B12 and folate deficiencies at six-month intervals. Other micronutrients such as copper, selenium, etc. are not routinely screened unless there is clinical suspicion of their deficiency. Concerning sexual dysfunction, the underlying mechanisms involved may be multifactorial and may include the hypogonadism itself and the association with the metabolic syndrome, which is known to lead to endothelial and erectile dysfunction [45]. Therefore, the mechanisms that explain the improvement in sexual function are closely related to an improvement in the hormonal and metabolic milieu after weight loss.

The limitations of our study include its observational nature, the number of patients included in the analysis, and the lack of more than one semen sample at each time in each patient. Recruitment of patients willing to repeatedly provide semen samples is often difficult, especially if serial samples are required during follow-up and if stringent recruitment criteria are applied. Unlike previous studies, sample size calculation was performed in order to assure enough power to detect clinically significant changes in the main semen parameters. However, as two different bariatric procedures were performed, this sample size did not provide sufficient power to detect if each technique may have differential effects. The main strength of this study is the collection of data throughout the phase of acute weight loss and until 18 months after surgery (when weight often stabilizes). This provides important information concerning the possible influence of active or stabilized weight loss on the parameters studied. Furthermore, we carefully selected patients to exclude any confounders that could affect gonadal function.

In conclusion, while weight loss through obesity surgery improves testosterone deficiency and sexual performance in obese men, its effects on fertility may be limited or even deleterious (at least in the short and midterm). Therefore, the presence of altered fertility in a patient with severe obesity should not be regarded as a trigger for obesity surgery. Further large-scale studies are needed to consolidate the findings provided by the present study and to explore the etiologies of impaired semen parameters in men with obesity.

## Figures and Tables

**Figure 1 jcm-11-05126-f001:**
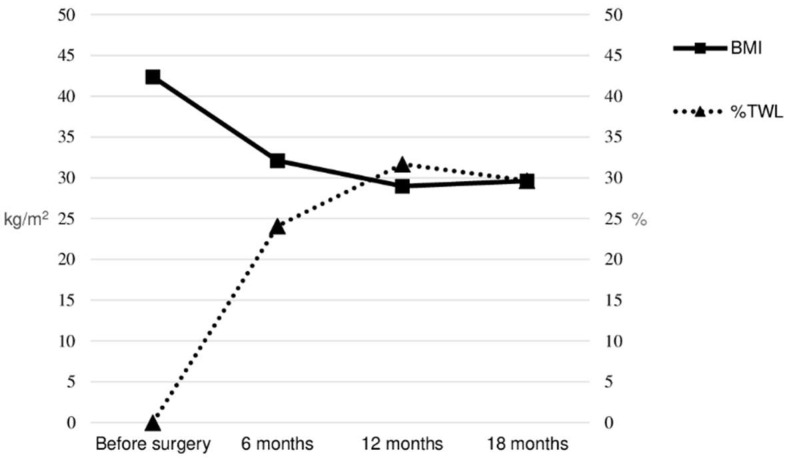
Evolution of BMI and %TWL during follow-up.

**Table 1 jcm-11-05126-t001:** Baseline and follow-up clinical and metabolic variables.

	Baseline (n = 12)	18 Months (n = 11)
Age (years)	45 ± 4.87	46.17 ± 4.86
BMI (kg/m^2^)	42.37 ± 4.44	29.6 ± 3.77 *
Diabetes mellitus, n (%)	3 (25)	2 (18.2)
Hypertension, n (%)	5 (41.7)	2 (18.2)
Dyslipidemia, n (%)	10 (83.3)	1 (9.1) *
OSA, n (%)	6 (50)	3 (27.3)
Hepatic steatosis † (%)	11 (9.7)	-
NAFLD fibrosis score	−1.34 ± 1.01	−1.69 ± 0.94
Fasting glycemia (mmol/L)	6.18 ± 1.47	5.71 ± 2.27
HbA1c (%)	5.75 ± 0.62	5.45 ± 0.97 *
Triglycerides (mmol/L)	3.66 ± 1.36	0.98 ± 0.41 *
Total cholesterol (mmol/L)	4.86 ± 0.91	4.44 ± 0.96 *
LDLc (mmol/L)	3.08 ± 0.77	2.63 ± 0.72 *
HDLc (mmol/L)	1.06 ± 0.16	1.36 ± 0.27 *

BMI, body mass index; OSA, obstructive sleep apnea. † Assessed by abdominal echography. Data are expressed as mean ± SD or n (%). * *p* < 0.05.

**Table 2 jcm-11-05126-t002:** Hormone and seminogram profile at baseline and during follow-up.

Hormone Variables	Before Surgery (n = 12)	6 Months (n = 11)	12 Months (n = 12)	18 Months (n = 11)
Total testosterone (nmol/L)	11.83 ± 4.99	21.19 ± 5.97 **	20.77 ± 6.61 **	22.91 ± 4.88 **
Free testosterone (nmol/L)	0.28 ± 0.06	0.42 ± 0.10 **	0.41 ± 0.15 *	0.41 ± 0.05 **
SHBG (nmol/L)	22.54 ± 12.10	39.13 ± 10.82 **	41.49 ± 12.55 **	41.30 ± 14.41 **
Estradiol (nmol/L)	0.11 ± 0.36	0.11 ± 0.38	0.11 ± 0.02	0.10 ± 0.05
Prolactin (mUI/L)	223.00 ± 142.3	142.972 ± 85.14	212.196 ± 156.09	213.22 ± 184.33
LH (UI/L)	2.91 ± 1.47	2.96 ± 0.97	2.93 ± 1.41	3.41 ± 1.23
MOSH (N (%))	4 (33.3%)	0 (0) *	0 (0) *	0 (0) *
FSH (UI/L)	4.28 ± 1.5	4.37 ± 1.7	4.53 ± 1.00	5.32 ± 1.68 *
**Semen parameters**	**Before Surgery (n = 12)**	**6 Months (n = 11)**	**12 Months (n = 12)**	**18 Months (n = 11)**
Semen volume (mL)	2.9 ± 1.75	2.59 ± 0.81	3.13 ± 1.85	3.16 ± 1.48
Semen pH	7.63 ± 0.25	7.66 ± 0.29	7.56 ± 0.18	7.57 ± 0.10
Sperm concentration (×10^6^/mL)	48.75 ± 34.43	40.21 ± 31.41	39.43 ± 29.26	44.85 ± 43.6
Total sperm count (×10^6^)	118.95 ± 79.91	110.12 ± 102.37	106.35 ± 105.46	157.08 ± 206.28
Leucocyte concentration (×10^6^/mL)	0.17 ± 0.39	0.00 ± 0.00	0.00 ± 0.00	0.31 ± 0.62
Sperm progressive motility (%)	38.66 ± 17.68	33.9 ± 20.62	31.12 ± 12.79	26 ± 14.71
Sperm with normal morphology (%)	5.83 ± 4.50	5.55 ± 4.54	3.14 ± 1.67	2.82 ± 2.08 *
Sperms with multiple defects (%)	43.83 ± 9.90	45.55 ± 8.65	49.57 ± 8.16 *	46.55 ± 8.26
Teratozoospermia Index (%)	1.55 ± 0.12	1.57 ± 0.11	1.60 ± 0.11	1.57 ± 0.10
Sperm DNA fragmentation (%)	25.27 ± 9.23	29.56 ± 13.48	34.43 ± 11.04	34.22 ± 13.89
Degraded sperm (%)	3.18 ± 3.74	1.67 ± 0.87	2.57 ± 0.79	4.22 ± 5.36
Patients with one or more abnormal semen parameters † (N (%))	9 (75)	6 (54.5)	7 (58.3)	9 (81.8)

† Includes volume, sperm concentration, progressive motility, normal morphology, Values are expressed as mean + SD or N(%). * *p* < 0.05 compared to pre-surgery values. ** *p* < 0.01 compared to pre-surgical values.

**Table 3 jcm-11-05126-t003:** Sexual function parameters.

Erectile Function Variables	Before Surgery (n = 12)	6 Months (n = 11)	12 Months (n = 12)	18 Months (n = 11)
Erectile function	23.08 ± 7.57	27.18 ± 6.19 *	26.82 ± 4.21	27 ± 4.26
Orgasmic function	9.25 ± 3.67	11.18 ± 3.06	11.18 ± 4.14	11.60 ± 2.011
Sexual desire	9.55 ± 1.036	9.18 ± 1.94	9.36 ± 1.28	9.90 ± 0.316
Intercourse satisfaction	6.75 ± 1.76	6.73 ± 1.27	6.36 ± 1.80	6.90 ± 1.96
Overall satisfaction	6.17 ± 2.08	7.64 ± 2.06	7.73 ± 2.53	7.70 ± 2.45

Values are expressed as mean + SD. * *p* < 0.05 compared to pre-surgery.

## Data Availability

Raw data were generated at Hospital de la Santa Creu i Sant Pau. Derived data supporting the findings of this study are available from the corresponding author (Antonio Pérez) on request.

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
