# Peer review of "Obesity Surgery Improves Hypogonadism and Sexual Function in Men without Effects in Sperm Quality"

_jcm, 2022, doi:10.3390/jcm11175126_

Round 1

Reviewer 1 Report

Obesity is the cause of many health complications and may affect fertility. It is a critical public health problem and it's important from the social perspective, but there is still little research into this correlation and pathophysiological mechanisms are not welThe present study aimedsent study was to analyze the effects of bariatric surgery on hormonal and sperm parameters, and on sexual function in men with severe obesity. 12 patients were studied at baseline and at 6, 12, and 18 months after obesity surgery. Correct methods were selected to achieve the research goals and proper statistical methods were used, and they are appropriate to test the hypothesis. Methods are clearly and precisely described, therefore it will be possible to repeat the study by other research groups in the future. 

The proportion of sperm cells with multiple defects increased from baseline to 12 months. There was a significant positive correlation between the increase in total testosterone and the percentage of total weight lost at 18 months. Regarding spermatic parameters, there was a tendency to decrease in the percentage of sperm cells with normal morphology after bariatric surgery, reaching statistical significance at 18 months, and an increase in the number of sperm with multiple defects (12 months). The number of morphologically normal sperm was found to decrease after bariatric surgery, which corresponded with the increase found in the number of sperm with multiple abnormalities. Data are very interesting, presented in a clear and understandable way. This finding is intriguing, considering the robust improvement in the hormonal milieu in patients witt obesity and secondary hypogonadism. Figures and tables are legible and properly prepared in accordance with the guidelines. 

The ethics statement is adequate. The study was approved by the Ethics Committee of Hospital de la Santa Creu i Sant Pau (protocol code IIBSP-GON-2013-74 June of 2013).

Questions:

Sample size calculation: what is the power of your study? please dascribe the assumptions, you base your sample size calculation on.

[53] Research shows that lifestyle modifications increase total and free testosterone concentrations in men after weight loss. Are there any data on sperm quantity and quality after lifestyle intervention? That would be an interesting comparison for your research group.

[71] You excluded patients with concomitant treatments that could disturb/interfere with sexual hormone concentrations or spermatic parameters - OK. But did you analyze what vitamins or supplements your patients were taking? As you noticed in the discussion - worsening of sperm characteristics after obesity surgery may be caused by the decreased absorption of nutrients and the presence of nutritional deficiencies. According to the recommendations, patients after bariatric surgeries should take multivitamin supplements, and, depending on the needs, also iron, B12. Have your patients taken such supplements? Such information should appear in the limitations, as it is an aspect that makes the study group even more heterogeneous.

Despite the above critiques, I consider the manuscript clear, relevant for the field and presented in a well-structured manner. Cited references are mostly recent publications and relevant, and there are no self-citations.

I would like to congratulate the authors of the interesting results, which are certainly an important report filling a certain gap in knowledge, but further research is needed to clarify the issues related to fertility disorders in obesity.

Reviewer 2 Report

Dear Editor,

Thanks for sending me the article to review. This study is a prospective longitudinal study to evaluate the effects of bariatric surgery on hormonal and sperm parameters and sexual performance of 12 severely obese men who underwent bariatric surgery. The topic is interesting. I have some questions and comments about the manuscript. My assessment is below:

1- This study appears to be a clinical trial and not a cohort observation. The formula for calculating the sample size is also for clinical trials.

If the study is observational, the sample size is very small and the results are unreliable.

This issue is very important and should be clarified.

2- In Table 1, the age value after 18 months is empty in the third column. This is also true for hepatic steatosis.

3- The topic of Table 3 should be sexual function parameters, instead of erectile function parameters.

 4- Where the authors have used the Wilcoxon test as indicated in the statistical analysis section.

Reviewer 3 Report

The aim of the present study was to analyze the effects of bariatric surgery on hormonal and sperm parameters, and on sexual function in men with severe obesity. The study is well-written and well-conducted, and limitations are well-stated and discussed.

Page 1, line 43: it may be worthwhile to cite this recently published review:  PMID 34829704

Line 87: ” Assessment of FSH, LH and estradiol was performed with….”

Line 139: data for SHBG is reported in the table. Hence, using “data not shown” is not correct.

Line 144: semen parameters is more correct than spermatic parameters.

Line 149: in order to be consistent, the dot “.” should be used for all decimals, instead of the comma “,”. Please revise this in the entire manuscript.

Are you sure that erectile function does not significantly improve at 18 moths? Please check it.

Please clarify if these men were already normozoospermic at the baseline, and if the percentage of oligo/astheno/terato patients change along the time.

Round 2

Reviewer 2 Report

Dear Authors,

Your study is interesting for readers, with good presentation quality, but I still do not agree with the sample size and asked the editor to consult an epidemiologist or a statistician.

Warmly Regards
